# Perfluorocarbons cause thrombocytopenia, changes in RBC morphology and death in a baboon model of systemic inflammation

Heather F. Pidcoke[1¤a], Wilfred Delacruz[2], Maryanne C. Herzig[1]*, Beverly S. Schaffer[1], Sahar T. Leazer[1¤b], Chriselda G. Fedyk[1], Robbie K. Montogomery[1¤c], Nicolas J. Prat[1¤d], Bijaya K. Parida[1], James K. Aden[1], Michael R. Scherer[1], Robert L. Reddick[3], Robert E. Shade[4], Andrew P. Cap[1]

1 Blood and Shock Resuscitation, U.S. Army Institute of Surgical Research, Fort Sam Houston, TX, United States of America, 2 Hematology-Oncology Service, San Antonio Military Medical Center, Fort Sam Houston, TX, United States of America, 3 Department of Pathology and Laboratory Medicine, University of Texas, Health Science Center, San Antonio, TX, United States of America, 4 Southwest National Primate Research Center, Texas Biomedical Research Institute, San Antonio, TX, United States of America

¤a Current address: Office of the Vice President of Research, Colorado State University, Fort Collins, CO, United States of America
¤b Current address: Department of Medicine, Uniformed Service University of the Health Sciences, Bethesda, MD, United States of America
¤c Current address: Department of Microbial Pathogenesis and Immunology, Texas A&M Institute for Neuroscience, Bryan, TX, United States of America
¤d Current address: Institut de Reserche Biomédicale Des Armées, Bretigny-sur-Orge, France
* maryanne.c.herzig.ctr@mail.mil

## Abstract

A perfluorocarbon (PFC) investigated for treatment of traumatic brain injury (TBI) delivers oxygen to support brain function, but causes transient thrombocytopenia. TBI can cause acute inflammation with resulting thrombocytopenia; an interaction between the PFC effects and TBI inflammation might exacerbate thrombocytopenia. Therefore, PFC effects on platelet (PLT) function and hemostasis in a lipopolysaccharide (LPS) model of inflammation in the baboon were studied. Animals were randomized to receive saline ±LPS, and ± one of two doses of PFC. PLT count, transmission electron microscopy, and microparticle populations were quantified at baseline (BL) and at 2, 24, 48, 72, and 96 hours; hemostatic parameters for aggregometry and for blood clotting were measured at baseline (BL) and days 3 and 4. Injection of vehicle and LPS caused thrombocytopenia within hours; PFCs caused delayed thrombocytopenia beginning 48 hours post-infusion. LPS+PFC produced a more prolonged PLT decline and decreased clot strength. LPS+PFC increased ADP-stimulated aggregation, but PFC alone did not. Microparticle abundance was greatest in the LPS+PFC groups. LPS+PFC caused diffuse microvascular hemorrhage and death in 2 of 5 baboons in the low dose LPS-PFC group and 2 of 2 in the high dose LPS-PFC group. Necropsy and histology suggested death was caused by shock associated with hemorrhage in multiple organs. Abnormal morphology of platelets and red blood cells were notable for PFC inclusions. In summary, PFC infusion caused clinically significant thrombocytopenia and exacerbated LPS-induced platelet activation. The interaction between these effects resulted in decreased hemostatic capacity, diffuse bleeding, shock and death.

**Data Availability Statement:** All relevant data are within the manuscript and its Supporting Information files.

**Funding:** Dr Robert Shade received funding from Oxygen Biotherapeutics, Inc. Dr. Andrew Cap received funding from US Army Medical Research and Materiel Command. The funders had no role in study design, data collection and analysis, decision to publish, or preparation of the manuscript.

**Competing interests:** The authors have declared that no competing interests exist.

## Introduction

Severe hemorrhage and traumatic brain injury (TBI) are leading causes of battlefield death. Clinicians must provide therapy that restores intravascular volume and facilitates oxygen delivery, improving tissue perfusion, without exacerbating coagulopathy and bleeding [1, 2]. Delayed restoration of tissue perfusion increases morbidity and mortality, however blood products are often unavailable in the pre-hospital environment, close to the point of injury. Perfluorocarbons (PFCs), composed of carbon and fluorine, are characterized by extremely high oxygen and carbon dioxide solubilities allowing for efficient oxygen/carbon dioxide exchange, oxygenation of hypoxic tissues and removal of carbon dioxide through passive diffusion critical in systems undergoing circulatory collapse [3]. Additionally, PFCs are thought to be biologically inert and pharmacologically inactive [4, 5]. Though immiscible in plasma, they can be dispensed in an emulsion form [5–7]. These properties suggested that PFCs might be an ideal blood substitute or supplement [8, 9].

PFC use has been investigated in the context of traumatic brain injury (TBI) where further brain tissue damage occurs as a result of secondary insults from hypoxia and metabolic derangements [10–12]. PFCs increased brain perfusion by 10% or more [3]. Pre-clinical studies in a cat model demonstrated PFCs have a protective effect on acute focal cerebral ischemia [12– 14], while studies in rat TBI models showed PFCs can increase brain tissue oxygen tension when administered soon after TBI [15]. At the cellular level, PFCs have been shown to improve mitochondrial oxygen consumption levels [16] and at the functional level, PFCs have been shown to improve cognitive recovery and reduce neuronal cell loss [17]. In both swine and rat models of spinal cord injury, PFCs were shown to be neuroprotective and to improve recovery [18–20].

Oxycyte, a third generation PFC, is an emulsion containing perfluoro (tert-butylcyclohexane). Oxycyte dissolves 3–4 times more oxygen than human hemoglobin can off-load under normal physiological conditions. The median size of PFC droplets is 40–50 times smaller than an erythrocyte, thus Oxycyte may be able to oxygenate tissues that erythrocytes cannot access due to vascular compromise or arteriole vasoconstriction [3, 7, 21–24].

Three clinical studies have been conducted in humans with Oxycyte; a total of 31 individuals have received Oxycyte at doses of 0.3 to 1.8 g/kg (0.5 to 3.0 mL/kg of a 60% w/v emulsion). The first study was conducted in healthy volunteers, the second in patients with repeat hip arthroplasty, and the third in patients with severe TBI (NCT00174980, NCT00908063). Decreased platelet count was observed in all patients who received Oxycyte infusion [24, 25]. Since acute inflammation is a well-known cause of platelet count decline, and severe TBI is characterized by inflammation, and activation of coagulation and platelets [26–28], concerns existed that the combined effect of Oxycyte and inflammation could exacerbate thrombocytopenia.

Baboons have historically been used as a model for human platelet function [29] and Oxygen Biotherapeutics, evaluated the maximum tolerated dose (MTD) of Oxycyte and its effects in baboons at the Texas Biomedical Research Institute (San Antonio, Texas). Baboons develop sepsis in response to lipopolysaccharides of gram-negative bacteria with manifestation of acute cytokine responses and thrombocytopenia [30]. Therefore a lipopolysaccharide (LPS) model of inflammation in the baboon was used in this study in conjunction with the PFC Oxycyte to provide a comprehensive evaluation of the PFC Oxycyte's effects on platelet function and hemostasis under conditions of exacerbated thrombocytopenia.

## Materials and methods

### Experimental animals

Protocol review and approval were obtained from the Institutional Animal Care and Use Committee (IACUC) of the Southwest National Primate Research Center (SNPRC) and the

US Army Medical Research and Materiel Command Office of Research Protections Animal Care and Use Review Office (ACURO). The study was conducted in compliance with the Animal Welfare Act, implementing Animal Welfare Regulations, and the principles of the Guide for the Care and Use of Laboratory Animals. Baboons were housed and monitored and all data collection was performed at the Texas Biomedical Research Institute.

Baboon age ranged from 6–14 yrs with the weight of the animals 30.8 kg±1.9 (SEM). Housing enclosures were compliant with standards as described in the USDA Animal Welfare Act (9 CFR, Parts 1, 2, and 3) and Guide for the Care and Use of Laboratory Animals (ILAR publication, National Academy Press, Washington DC, 2011). Animals were not commingled during the course of this study. Single cage housing was necessary in order to monitor adequacy of food intake and to obtain blood samples during the course of the study.

Animals were removed from group housing to single housing one week prior to their procedures to allow animals to acclimate and to facilitate monitoring of baseline food intake. A complete physical examination under ketamine-induced sedation was performed by a staff veterinarian on all animals. Clinical pathology included hematology and coagulation parameters. Animals were confirmed free of parasites and tuberculosis before study arm assignment which used a computer-generated process of randomization based on body weight to control bias. The study animals were observed at least twice daily illness or distress; findings were promptly reported to a staff veterinarian and the principal investigator. Feed (Monkey Diet 15%, LabDiet 5LE0) was provided on a routine feeding schedule in amounts appropriate for the size and age of the animals. Tap water was available ad libitum. Prior to sedation and anesthesia for study procedures, food and water were held to minimize the risk of aspiration and were offered again after recovery from anesthetic.

There were six randomized groups: saline-saline control (SAL-SAL), saline-PFC3 (SAL-PFC3), saline-PFC12 (SAL-PFC12), LPS-saline (LPS-SAL), LPS-PFC3, and LPS-PFC12. The LPS dose was 0.3 mg/kg, while the PFC3 and PFC12 doses were 3 ml/kg and 12 ml/kg, respectively, for Oxycyte delivery of 1.8 mg/kg and 7.2 mg/kg. Individual group animals were randomly studied. Study groups and drug dosages are shown in Table 1 and Fig 1.

## Surgical procedures

On the morning of Study Day 0 (day of dosing), a baseline blood sample was collected followed by an intravenous infusion with LPS (0.3 mg/kg in 20 ml saline) or LPS vehicle (saline) for 30 min at an infusion rate of 0.67 mL/minute via the saphenous or other appropriate peripheral vein. Approximately 30 minutes after completion of the LPS infusion, PFC or vehicle (saline) was infused at a constant infusion rate that delivered the intended dose over 30 minutes via the saphenous or other appropriate peripheral vein. The volume administered to each animal was calculated and adjusted based on the body weight collected on the day of administration. Blood samples were collected at 1, 1.5, 3, 7, 25, 49, 73, 97, and 121 hours after starting the LPS infusion. Baboons were sedated with ketamine and then maintained on isoflurane anesthesia through the 1.5 hour blood sample time point. All other blood samples were collected under ketamine or medetomidine sedation at the discretion of the Responsible Veterinarian. PFC

**Table 1. Drug dosage and study groups.**

| | | GROUP | | | | | |
| | | SAL-SAL | LPS-SAL | SAL-PFC3 | LPS-PFC3 | SAL-PFC12 | LPS-PFC12 |
| --- | --- | --- | --- | --- | --- | --- | --- |
| LPS | DOSE | ---- | 0.3 mg/kg | ---- | 0.3 mg/kg | ---- | 0.3 mg/kg |
| PFC | VOL | ---- | ---- | 3.0 ml/kg | 3.0 ml/kg | 12.0 ml/kg | 12.0 ml/kg |

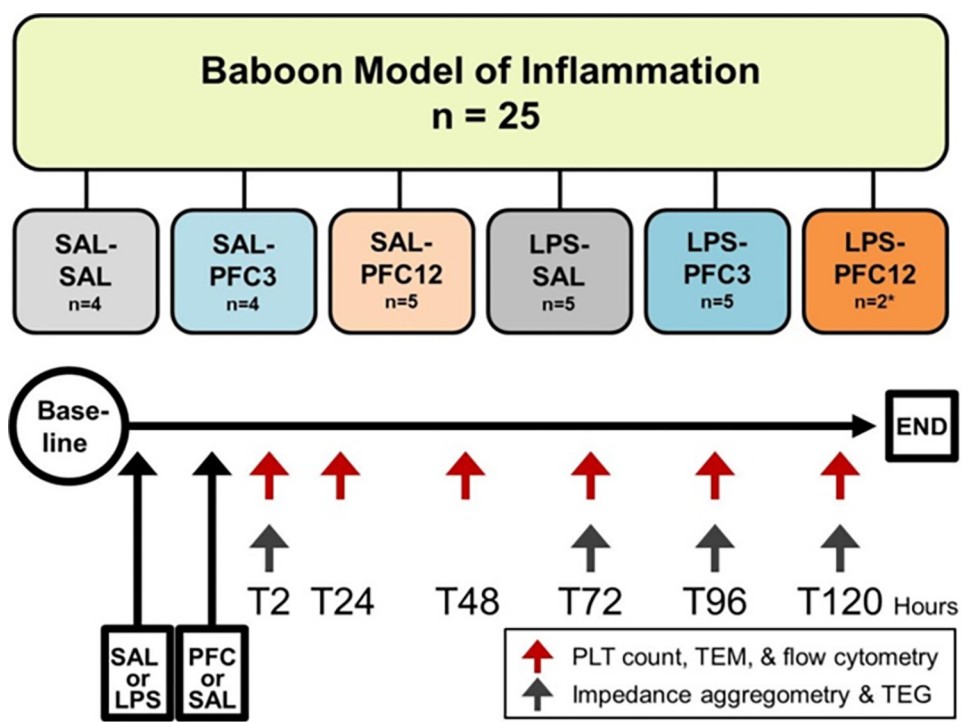

**Fig 1. Study groups and experimental schema.**

(Oxycyte®, 60% w/v perfluoro (t-butylcyclohexane), Tenax Therapeutics, formerly Oxygen Biotherapeutics Inc., Morrisville, NC) was supplied as a sterile, ready to administer emulsion. *E. coli* LPS was obtained as an irradiated sterile powder (Sigma-Aldrich L2630) and reconstituted into a stock solution with sterile saline (sodium chloride injection, USP [0.9%]), and stored frozen (-20°C). Aliquots were thawed on the day of administration and transferred to a capped syringe for dose administration.

The study animals were observed at least twice daily for signs of illness or distress.

Animals with pronounced effects (e.g., severe emesis, convulsions) were removed from the study and treated with supportive therapy. The responsible veterinarian immediately acted at their discretion to alleviate pain or distress by appropriate means, including blood transfusions. Baboons displaying severe petechia, difficulty breathing, or severe anemia were humanely euthanized with Fatal-Plus, a pentobarbital sodium injection.

### Blood analyses

Platelet count, transmission electron microscopy (TEM), and flow cytometry were performed on blood samples at baseline (BL) and at 2, 24, 48, 72, 96, and 120 hours (T2, T24, T48, T72, T96, T120) after PFC infusion. A complete blood count was performed on EDTA anti-coagulated blood using a Unical 800 DxH analyzer (Beckman Coulter). Prothrombin time (PT), activated partial thromboplastin time (PTT) and fibrinogen levels were analyzed using citrated plasma samples on an ACL8000 (Instrumentation Laboratory). The following cytokine levels were analyzed by ELISA on EDTA anti-coagulated plasma stored at -70°C: interleukin-6 (IL-6), interleukin 1 receptor antagonist (IL-1ra; R&D Systems, Minneapolis, MN), tumor necrosis factor-alpha (TNF-α), macrophage inflammatory protein 1a and 1b (MIP-1a; MIP-1b; Invitrogen, ThermoFisher, Waltham, MA). P-selectin and soluble CD41levels were determined by

ELISA on citrated plasma after storage at -70˚C (human sCD40L Platinum ELISA and human sP-selectin Platinum ELISA, Affymetrix, eBioscience, ThermoFisher, Waltham, MA). Microparticles (MP) were quantified and characterized by flow cytometry; platelet and red blood cell morphology were examined via TEM. Impedance aggregometry (Multiplate 5.0 Analyzer, Dynabyte Medical, Munich, Germany) and thromboelastography (TEG; Haemonetics Corporation, Braintree, MA) were quantified at BL and at T2, T72, and T96 (hours) as diagrammed in Fig 1.

## Microparticle (MP) analysis

Citrated whole blood was centrifuged at 3000xg for 10 minutes; supernatant was removed and re-centrifuged as above to produce Platelet Free Plasma (PFP). PFP was stored on ice and immediately transported to the US Army Institute of Surgical Research (USAISR) for processing. 10µl PFP was added to an antibody cocktail containing the following antibodies: APC labeled CD41a, clone HIP8; PE labeled CD62P, clone AC1.2; and V450 labeled CD45, clone D058-1283 (all from BD Biosciences, San Jose, California, USA,); PerCP-Cy5.5 labeled CD144 (Santa Cruz Biotechnology, Dallas, Texas, USA,) and 100 nM FITC labeled lactadherin (Haematologic Technologies, Inc., Essex Junction, Vermont, USA,). Isotype controls were used for non-specific fluorescence: PE mouse IgG1k, clone x40; PerCP isotype, mouse igG 1K, clone MOPC-21; APC isotype, Mouse igG 1K, clone MOPC-21; V450 isotype control clone MOPC-21. Color compensation was performed to account for fluorescence spectral overlap. After 30 minute incubation in the dark, on ice, 1 ml of 0.1µ filtered Hanks Balanced Salt Solution (HBSS) without calcium, magnesium or Phenol Red (Gibco/ Life Technologies, Grand Island, New York) was added. Analysis was performed on a BD FacsCantoII (RUO) equipped with 488 nm, 633 nm and 405 nm lasers and a FSC-PMT detector capable of detecting particles in the 200 nm range. The instrument was calibrated for microparticle size with polystyrene beads (201, 390, 505, 794 and 990 nm). Calibration curves were generated by plotting the log of the FSC-PMT median versus the bead size; 50,000 events were collected in the microparticle gate (100–1000 nm). Absolute concentration of microparticles was determined using BD TruCount absolute counting tubes [31] (BD Biosciences, San Jose, California, USA); 5µl PFP was added to 1 ml of 0.1µm filtered HBSS, well mixed and analyzed. Analysis was stopped after 1,000 events in the TruCount bead gate, and the absolute concentration of microparticles was calculated according to the following formula: $\frac{events\ in\ MP\ gate}{events\ in\ TruCount\ bead\ gate} X \frac{beads\ per\ test}{test\ volume}$.

## TEG

Fresh whole blood at T0, T2, T72 and T96 time points was placed in a 1.8 ml vacutainer containing 3.2% citrate, allowed to rest 15 min at room temperature, after which thromboelastography analysis of blood coagulation was performed on a TEG 5000 Thromboelastograph Hemostasis Analyzer System (Haemonetics Corporation, Braintree, MA) according to manufacturers' instructions [32]. Specifically, 1 ml of carefully mixed blood (slow inversion four times) was added to a kaolin containing tube (Haemonetics). The kaolin-blood mix (340 µl) was added to the test cup containing 20 µl of 0.2 M calcium chloride; the TEG assay was monitored for 3h duration. Parameters reported include R (lag time to start of clot), and MA (maximum amplitude of clot).

## Platelet aggregometry

Fresh whole blood at T0, T2, T72 and T96 time points was placed in a 2.7 ml vacutainer containing 3.2% citrate, allowed to rest 30 min at room temperature, after which platelet

aggregometry was measured on a Multiplate 5.0 Analyzer (Diapharma, West Chester, Ohio) according to manufacturer's instructions [32]. Specifically, 300 μl of carefully mixed whole blood was incubated with 300 μl saline, or saline with 2 mM calcium chloride for 3 minutes in a test cell at 37˚C with continuous mixing; agonist was then added and the aggregometry determined over a six minute assay. Agonists and their final concentrations were: ADP, 6.5 μM; collagen, 3.2 μg/ml; thrombin receptor activating peptide (TRAP), 32 μM; arachidonic acid (ASPI), 0.5 mM and ristocetin, 0.77 mg/ml.

## TEM processing and analysis

At each time point, two 0.5 ml blood samples were dispensed into BD Microtainer™ MAP microtubes containing 1.0 mg K2EDTA. After mixing, the blood was pooled and centrifuged at 1000xg for 20 min at room temperature. Plasma was removed, leaving the buffy coat and packed red blood cells undisturbed. TEM buffer [33] replaced the original volume of plasma over the buffy coat/ packed red blood cells. Samples were stored at room temperature before transportation to the Pathology Electron Microscopy Facility at the University of Texas Health Science Center–San Antonio (UTHSCSA) for processing. Specimens were sectioned in an Ultracut Microtome and stained with uranyl acetate and Reynold's Lead Citrate before imaging.

## Tissue handling

Necropsies were performed by a registered veterinary pathologist within 1 h of death. Tissue samples (<4mm diameter) of liver, kidney, spleen, adrenal glands, lung and bone marrow were taken and placed in TEM buffer and/or 10% phosphate buffered formalin solution and stored at room temperature.

## Statistical analysis

Statistical analysis was by ANOVA, followed by pairwise comparisons and a Bonferroni adjustment. Analysis was performed using either SAS (SAS Institute, Inc., Cary, North Carolina) or SPSS software (SPSS 17 for Windows, SPSS, Chicago, IL).

## Data sharing statement

Data analyzed is found in an excel datasheet in S1 Data.

## Results

Animal numbers within study arms were as follows: SAL-SAL, n = 4; SAL-PFC3, n = 5; SAL-PFC12, n = 5; LPS-SAL, n = 4; LPS-PFC3, n = 5; and LPS-PFC12, n = 2. Randomization to the LPS-PFC12 group was halted due to early deaths. Data is shown however this group was not included in the statistical analyses due to insufficient data for analysis.

## Inflammation induction

LPS is known to elicit acute responses in the baboon in interleukin 6 (IL-6), tumor necrosis factor alpha (TNFa), macrophage inflammatory proteins 1alpha and 1 beta (MIP1a, MIP1b) as well as interleukin 1 receptor antagonist (IL-1Ra) [30, 34]. Infusion of LPS caused the expected inflammatory response in all LPS groups with a rapid increase in IL-6 and TNF-a (Fig 2), and in IL-1Ra, MIP-1a, and MIP-1b (S1 Fig in S1 File). A modest, yet significant, change over baseline is seen for IL-6 for the vehicle control groups of SAL-SAL, SAL-PFC3 and SAL-PFC12. In contrast, there is a four to five-fold order of magnitude difference in the increase in TNF-α and IL-6 levels of the baboons in the LPS inflammatory model groups. With the addition of

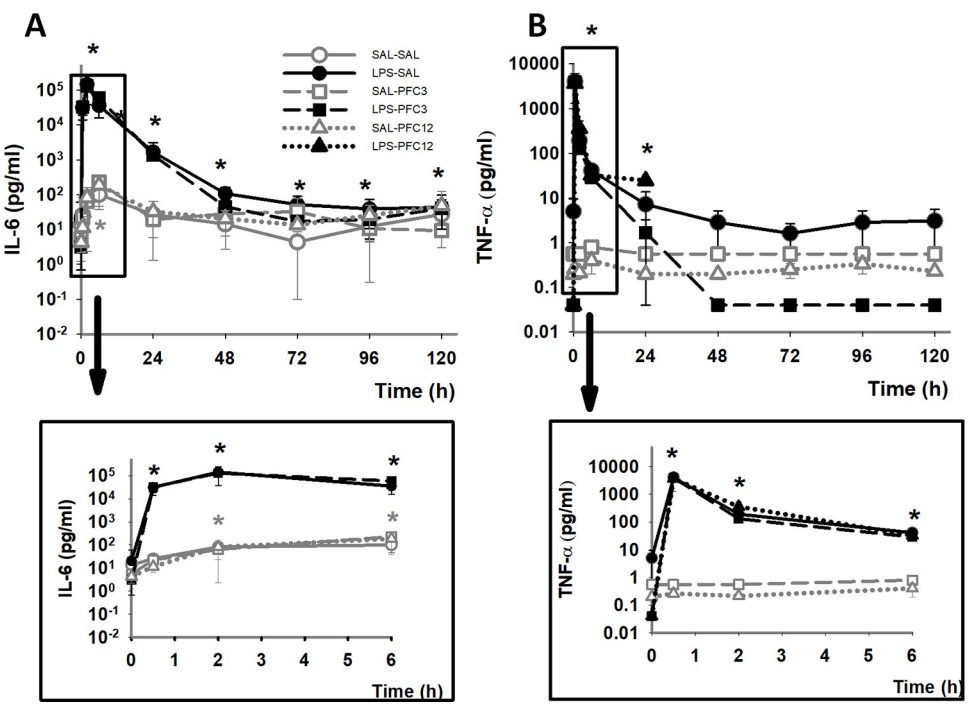

**Fig 2. LPS induces an immediate inflammatory response; PFC does not.** A. Baboons treated with an infusion of LPS in the presence or absence of PFC show an immediate increase in TNFα (A) and IL-6 (B). Upper panels show entire time course. Lower panels show early time points on an expanded scale. Open or closed symbols depict initial infusion conditions with open for saline and closed for LPS. Graphs of initial infusions with saline are in grey. Symbol type depicts second infusion with squares, circles and triangles for saline, PFC3 and PFC12, respectively. Error bars depict SEM. Data that are significantly different, p<0.05, are shown compared to: baseline, by *. Both treatment group and time show significant differences with an interaction between both treatment group and time.

PFC, there is no change in the initial response of the LPS groups; there is a significant drop in the TNFα levels by 48 h after induction compared to the LPS-SAL, however baseline levels for LPS-PFC3 were also significantly lower than the LPS-SAL control.

## Platelet numbers

Compared to the saline control group, the infusion of LPS caused a transient decline in PLT count within six hours with a full recovery after day 4 (p<0.05; Fig 3A). At its nadir, PLT count in the LPS-SAL group was >50% of baseline. Infusion of PFC with SAL both at 3 ml/kg and 12 ml/kg doses resulted in a delayed but persistent decline in PLT count compared to the SAL-SAL control group that started around day 2 (p<0.05; Fig 3B). This nadir was <50% of baseline at day 5. The combination of LPS and PFC resulted in an additive decline in PLT counts at all time points, occurred within two hours of infusion and persisted for more than five days (p<0.05). At the nadir, PLT count was <15% of baseline values. The thrombocytopenia observed correlates with decreased mean platelet volume (MPV) followed by an increasing MPV, indicative of younger platelet populations (S3 Fig in S1 File).

## Platelet aggregation

In aggregometry response to ADP agonist stimulation, there was an early drop for the LPS-SAL and LPS-PFC groups seen by 2h and a late drop for SAL-PFC groups, seen at 72 h (Fig 4A). Whereas SAL-PFC infusion inhibited platelet aggregation (p<0.05, SAL-PFC3), LPS infusion enhanced platelet aggregation by 72 h compared to SAL-SAL (p<0.05 for LPS-SAL).

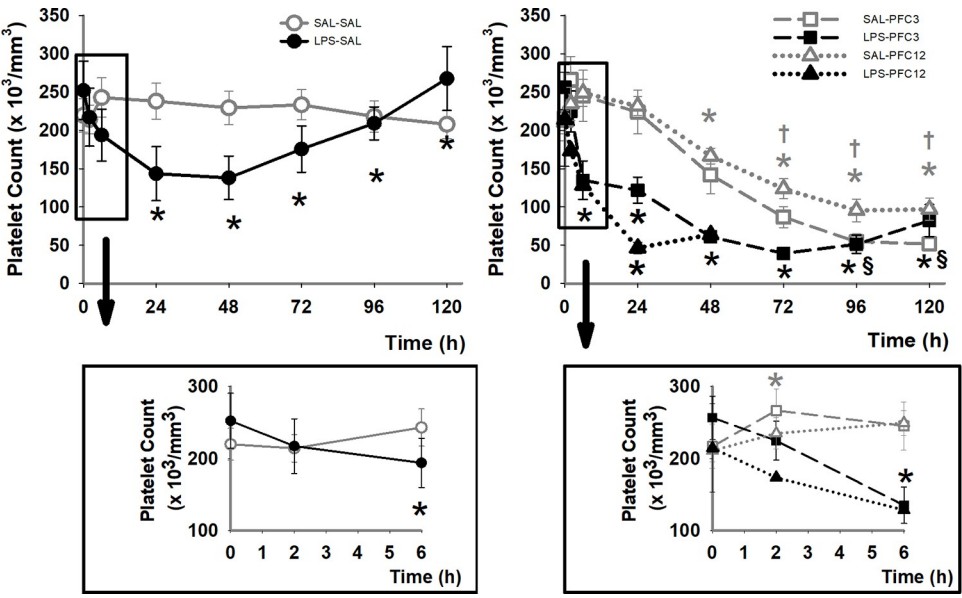

**Fig 3. PFC addition exacerbates LPS induced thrombocytopenia.** A. Baboon platelet numbers show a transient decrease in response to LPS with recovery by 120 h. B. Infusion of PFC in the presence or absence of LPS results in a persistent decline in PLT count. Upper panels show entire time course. Lower panels show early time points on an expanded scale. Open or closed symbols depict initial infusion conditions with open for saline and closed for LPS. Graphs of initial infusions with saline are in grey. Symbol type depicts second infusion with squares, circles and triangles for saline, PFC3 and PFC12, respectively. Error bars depict SEM. Data that are significantly different, $p < 0.05$, are shown compared to: baseline, by *; to SAL-SAL (SAL-PFC groups only), by †; and to LPS-SAL (LPS-PFC groups only), by §. Both treatment group and time show significant differences with an interaction between both treatment group and time.

The combination of LPS and PFC caused platelet activation similar to observations in the LPS-SAL group. There is a pronounced thrombocytopenia with PFC with a 75–80% drop in platelet number by 72h (see Fig 3) and as thrombocytopenia may affect aggregometry results, masking PFC and LPS effects on agonist response, the data was normalized to PLT count at each time point (Fig 4B). The effects of LPS and PFC on PLT aggregometry were preserved even after normalization to PLT count. Similar responses were seen for the agonists ASPI and TRAP (Fig 4C and 4D), Overall, the combination of LPS and PFC caused an initial suppression of platelet aggregation function followed by activation and hyper-reactivity at 72 hours.

## TEG

Consistent with platelet activation seen by aggregometry, thromboelastography demonstrated faster clot initiation (R) that was sustained compared to baseline for the SAL-PFC12 and the LPS-PFC3 groups after 72 hours (Fig 5A). Clot strength, measured by maximum amplitude of the clot (MA, a function of platelet and fibrinogen effects), was minimally decreased in the SAL-PFC12 group at day 4 ($p < 0.05$; Fig 5B). MA paradoxically increased over time ($p < 0.05$) with LPS-SAL, however, the combination of LPS and PFC decreased TEG MA values at all time points after infusion ($p < 0.05$). LPS and PFC3 interact in such a way that the LPS effect of increasing MA is more than reversed.

## Microparticle analysis

MP counts increased over time in the PFC3 group (Fig 6A). Other effects may have been masked by effects of the lipid particles in the PFC emulsion on flow cytometry. The CD41

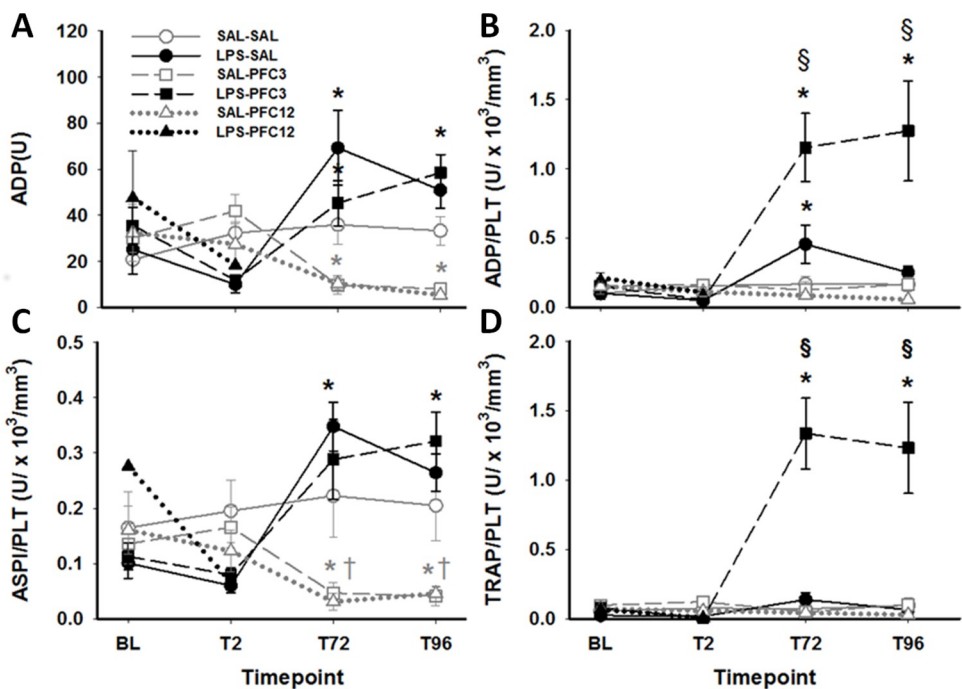

**Fig 4. Platelets were inhibited by PFC administration and activated in response to LPS.** A. Platelet aggregation in response to ADP. B. ADP aggregation normalized to platelet count C. ASPI aggregation normalized to platelet count. D. TRAP aggregation normalized to platelet count. Open or closed symbols depict initial infusion conditions with open for saline and closed for LPS. Symbol type depicts second infusion treatment with squares, circles and triangles for saline, PFC3 and PFC12, respectively. Data that are significantly different, $p<0.05$, are shown compared to: baseline, by *; to SAL-SAL (SAL-PFC groups only), by †; and to LPS-SAL (LPS-PFC groups only), by §.

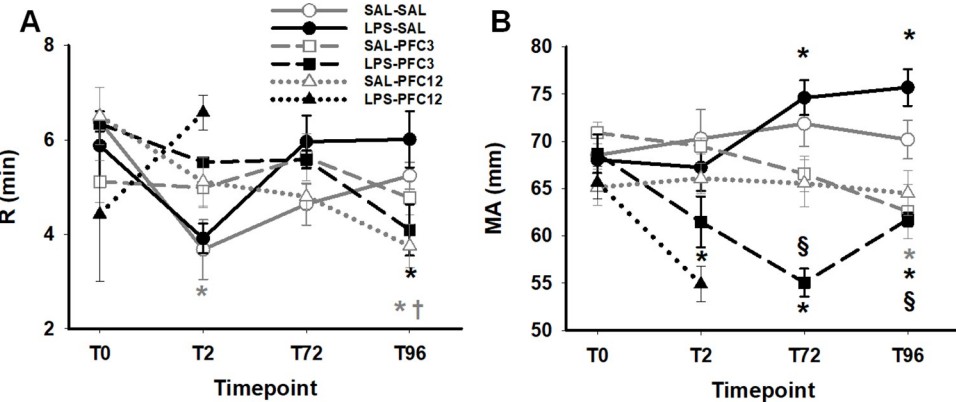

**Fig 5. The combination of LPS and PFC caused faster clot initiation and decreased clot strength.** A. Clot initiation is accelerated with PFC by 4 days. Both SAL-PFC12 and LPS-PFC3 are significantly decreased from BL levels (*, $p < .05$); SAL-PFC12 is significantly different from SAL-SAL control (†, $p < .05$). B. Changes in clot strength over time are shown. Significant changes from BL for LPS groups are depicted by * ($p < .05$); significant differences between LPS-SAL and LPS-PFC3 at time pts are shown by § ($p < .05$). Comparing groups, there is a significant decline in TEG MA in LPS-PFC3 group compared to LPS-SAL group at T72 and T96 time points ($p<0.05$). By RM ANOVA, the time effects were obscured by interaction with group: SAL-SAL, $p = NS$; LPS-SAL, MA increases with time; T0 v T72 and T96, $p\leq0.05$; SAL-PFC: MA drops @ T96, $p\leq0.05$; LPS-PFC: T0 v all others, $p\leq0.05$ with a recovery at T96, ($p\leq0.05$ compared to T72); SAL-PFC: no significant change, $p = NS$. In group effects, LPS-SAL was significantly different from LPS-PFC3 and SAL-PFC3 was significantly different from LPS-PFC3.

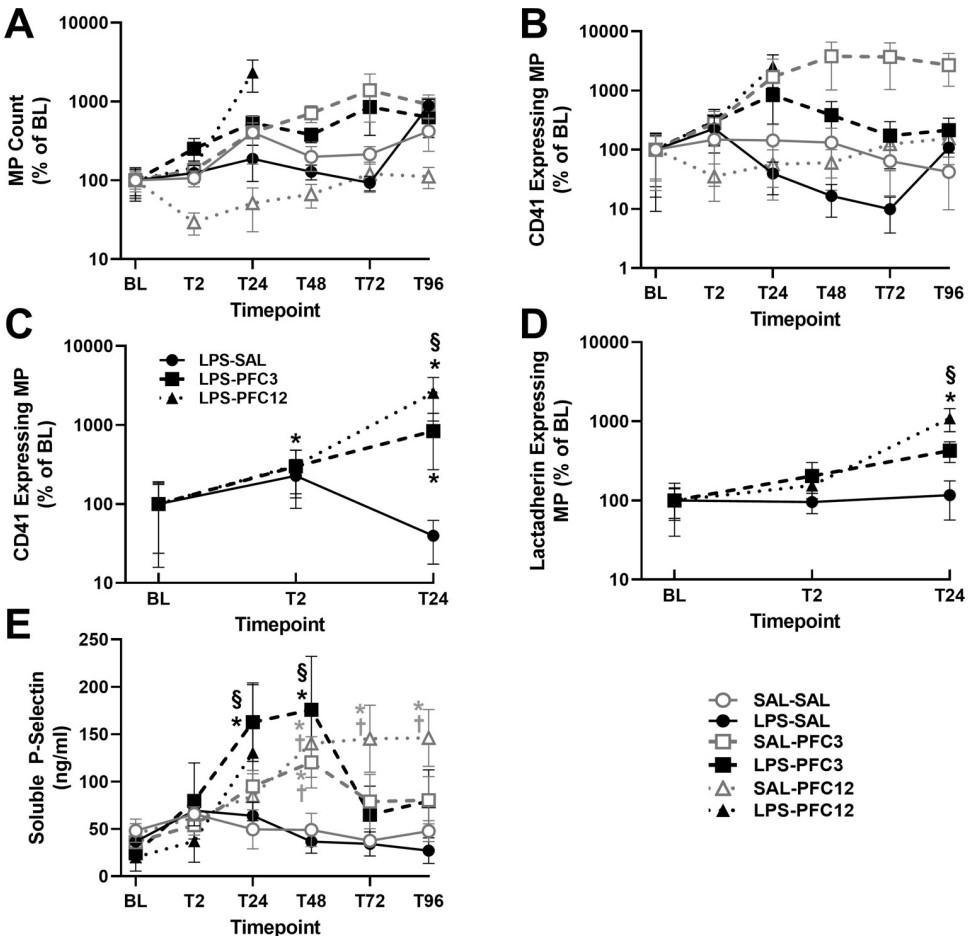

**Fig 6. PFC elevates microparticles.** A. Total microparticles show a significant group*time effect (p<0.05) with SAL-PFC3 and LPS-PFC3 elevated. B. Activated platelet MP containing GIIba (CD41 positivity). No significant group or time effects are seen. C. LPS 24h Subset of GIIb activated PLT MP. D. LPS 24h subset of lactadherin activated PLT MP. Graphs depict a log scale of increase over BL. E. Elisa of soluble P-selectin reveal LPS-PFC3 shows significant difference from baseline (*) and from LPS-SAL (§). SAL-PFC3 and SAL-PFC12 significant differences from SAL-SAL are shown with (†). Significance set at p<0.05.

positive platelet-generated MP were not significantly changed over time or by treatment (Fig 6B). However, GP IIb-expressing MP (CD41 positive) and lactadherin-binding MP (phosphatidyl serine positive) were significantly elevated in LPS-PFC groups within the first 24 hours, but not in the LPS-SAL groups (Fig 6C and 6D). Possible platelet fragmentation was also measured by soluble P-selectin ELISA (Fig 6E). Mirroring the platelet graphs, LPS-PFC3 significantly increased soluble P-selectin with a maximum by day 2 before returning to baseline; SAL-PFC3 and SAL-PFC 12 caused a significant delayed increase in soluble P-Selectin.

## Survival data

Baboons that tolerated the LPS and PFC combination tolerated it well over the duration of the study. The maximum weight loss in any group of surviving animals was 3.1% (Fig 7A). All animals in the control, LPS-SAL, and SAL-PFC3 groups survived (Fig 7B). There were four early deaths, 2/2 in the LPS-PFC12 group and 2/5 in the LPS-PFC3 group. Diffuse bleeding was noted in all four animals. One late death was noted in SAL-PFC12 group presumed to be

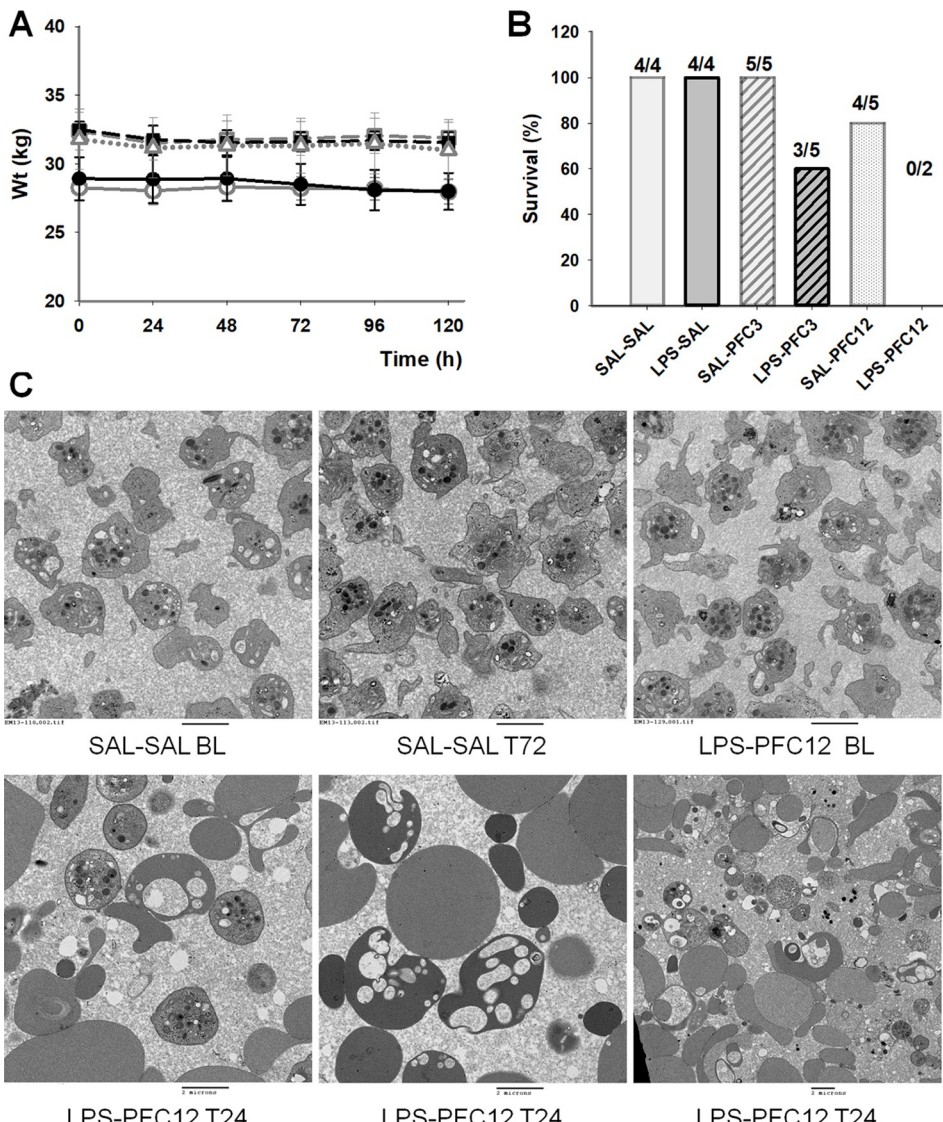

**Fig 7. Inflammation and perfluorocarbon would have been considered well tolerated except for study deaths; transmission electron microscopy capture changes in blood morphology.** A. Weight loss over study duration is shown. B. Mortality numbers from study. C. TEM images. Representative images from the control SAL-SAL group at BL and T72. Similarly, a representative BL image for the LPS-PFC12 group as well as representative images from the last available timepoint of this group, T24 are shown. The bar represents 2 microns. The first 5 Images were determined at 80kV with a direct magnification of 10000x and a print magnification of 15500 x 8 in with the final image at 5000x with a print magnification of 7760x8in.

secondary to femoral venipuncture site hemorrhage. Signs of inflammation consistent with infection were noted near the non-healing wound as well as diffuse vacuolated macrophages in multiple organs. Baseline PLT count measurements for three of the five animals that died were in the low normal range.

## TEM

Transmission Electron Microscopy (TEM) studies of platelets showed platelet activation with fibrin formation between baseline and T96 in the groups who received SAL-LPS, SAL-PFC3/

PFC12, and LPS-PFC3/PFC12 (Fig 7C; S1 Data). TEM of red blood cells (RBC) shows no changes in morphology between the control group, SAL-LPS and SAL-PFC groups. However, in the LPS-PFC3 and LPS-PFC12 groups, RBC fragmentation with vacuolization is identified. These changes are markedly pronounced at 24 hours.

## Histopathology

Necropsy performed on one animal from the SAL-PFC12 group found multifocal myocarditis and mild interstitial kidney nephritis with minimal neutrophilic tubulitis. Histiocytes with clear vacuoles were identified in the spleen, lung, liver, adrenal gland and bone marrow. One of the two animals that expired in the LPS-PFC3 group demonstrated diffuse intestinal congestion and hemorrhage, consistent with endotoxic shock. Diffuse severe acute nephrosis with necrosis, acute multifocal hepatitis, diffuse lung congestion and hemorrhagic necrosis of the spleen and adrenals were noted in the other. The two animals from the LPS-PFC12 group that expired displayed lesions suggestive of disseminated intravascular coagulation (DIC) and sepsis; findings included necrotizing splenitis, tubular necrotizing nephropathy with thrombi, congestive hepatitis, cortical adrenal necrosis and bacterial colonization of the lung and bone marrow. TEM studies of tissue samples from this group revealed sinusoidal accumulation of foamy cells with microsteatosis and oncocytic liver changes. In the kidney, glomerular microvesicular material and tubular pigment deposits were identified along with mitochondrial swelling and degeneration (S1 Data).

## Discussion

Third generation PFCs have high gas solubility and thus can deliver $O_2$ to ischemic tissues, potentially reducing morbidity and mortality. PFC infusion increased oxygen consumption in both human and animal studies, and improved cognition with decreased neuronal cell loss in a rat TBI model. Oxygen support with PFCs resulted in smaller infarct size and fewer deficits in a rat model of embolism, as well as improved spinal cord injury recovery in a swine model of decompression sickness. A known side effect is thrombocytopenia, a condition also seen in critically injured patients with inflammatory changes. To investigate the *in vivo* effects of a third generation PFC in the setting of inflammation, doses equivalent to those previously studied were evaluated in a baboon model of inflammation.

As expected, PFC caused delayed, persistent thrombocytopenia beginning 48 h post infusion with nadirs ≤50% of baseline and recovery delayed for 5 days or more. LPS caused more immediate, but transient decreases in platelet count. In combination, the effects were additive (Fig 3). The mechanism appears to be both platelet activation and clearance; PFC increased aggregation responses in the presence of LPS, which were not seen with SAL-PFC infusion. Increases in microparticles expressing constitutive platelet markers (Fig 6) and elevation in soluble P-Selectin support platelet clearance with LPS-PFC. Changes in global hemostasis, namely faster clot initiation in the LPS-PFC and SAL-PFC12 groups and decreased clot strength, were consistent with coagulation system activation and platelet consumption. The non-survivor rate was 20%, which is a surprising finding given that no animal deaths were expected in this survival study. Although pathology reports were consistent with shock, suggesting an excessive LPS dose, all early deaths occurred in the high PFC dose combined with LPS; there were no deaths in the LPS-SAL controls. The histology was consistent with shock associated with hemorrhage in multiple organs. Additionally, PFC treatment may have contributed to the delayed death due to femoral artery bleed in one SAL-PFC12 animal. PFC treatment appears to have potentiated the effects of LPS, rendering a normally tolerated dose lethal.

Both PLT and red blood cell morphology from animals treated with PFCs were abnormal as demonstrated by TEM. PFCs accumulate in these cells and distort cellular architecture which may adversely affect function and clearance. Direct effects on white blood cells and macrophages are unknown, but the autopsy findings in the late deaths are suggestive of impaired wound healing.

Limitations in this study are that actual infection and sepsis were not studied and the time dependency between PFC administration and LPS administration was not investigated. This PFC formulation is cleared from the circulation by macrophage phagocytosis of the particles approximately a week after administration and localizes to the liver and lungs. Elimination is through exhalation [35], a process that takes many months depending on the administered dose. While no toxic metabolites are formed, there is long-term stimulation of macrophages; the effects in the presence of inflammation, infection, or sepsis are not well understood. However, enhancement of inflammatory signaling molecules was noted in a rat model of partial liquid ventilation in the presence of LPS [34]. Knowing whether LPS administration after sequestration of PFC within liver macrophages is similarly harmful is relevant to treating TBI. Patients with severe TBI often require extended intensive care and are at risk of septic or infectious events during their hospital course. It is unknown if PFC treatment followed by exposure to LPS would decrease survival, or whether PFC sequestration within macrophage vacuoles would be protective. The known effect on macrophage stimulation suggests that at least some of the harmful effects, particularly in the lungs, may be due to an enhanced release of inflammatory mediators in response to LPS. While further study is required, the late death of a baboon exposed to PFC alone due to chronic inflammation and poor wound healing is concerning.

Expert Opinion: This study emphasizes the critical importance of attempting to anticipate clinical safety issues in the interrelationships of trauma, hemorrhage, and sepsis and then devising studies to test critical physiologic pathways during pre-clinical product development. This study identified thrombocytopenia and lethal hemorrhage as an adverse safety concern when the PFC blood substitute, Oxycyte, was administered in a baboon non-human primate model in the setting of LPS-induced inflammation. The findings obtained in this clinically relevant model contributed to termination of development efforts and avoided potential serious adverse events in human subjects.

In conclusion, third generation perfluorocarbon (Oxycyte) infusion caused clinically significant thrombocytopenia and platelet dysfunction, which was exacerbated by LPS-induced platelet activation. The interaction between inflammation and PFC resulted in abnormal cellular morphology, decreased hemostatic capacity, diffuse bleeding, shock and death.

## Supporting information

**S1 File. Supplementary figures.**
(PDF)

**S1 Data.**
(XLSX)

## Acknowledgments

The authors thank Stephanie Anderson (Oxygen Biotherapeutics Inc) for support and design of the experiments. Additionally, they would like to thank Kerri Stewart, RN and Hiram Garrastazu, BS for expert technical help.

## Author Contributions

**Conceptualization:** Heather F. Pidcoke, Robert E. Shade, Andrew P. Cap.

**Data curation:** Maryanne C. Herzig, Beverly S. Schaffer, Chriselda G. Fedyk, Robbie K. Montogomery, Nicolas J. Prat, Bijaya K. Parida, Robert E. Shade.

**Formal analysis:** Heather F. Pidcoke, Maryanne C. Herzig, Beverly S. Schaffer, Sahar T. Leazer, Chriselda G. Fedyk, Robbie K. Montogomery, Nicolas J. Prat, Bijaya K. Parida, James K. Aden, Robert L. Reddick, Robert E. Shade.

**Funding acquisition:** Heather F. Pidcoke, Robert E. Shade, Andrew P. Cap.

**Investigation:** Heather F. Pidcoke, Maryanne C. Herzig, Beverly S. Schaffer, Robbie K. Montogomery, Nicolas J. Prat, Bijaya K. Parida, Robert E. Shade, Andrew P. Cap.

**Methodology:** Heather F. Pidcoke, Chriselda G. Fedyk, Robbie K. Montogomery, Nicolas J. Prat, Robert E. Shade, Andrew P. Cap.

**Project administration:** Heather F. Pidcoke, Chriselda G. Fedyk, Michael R. Scherer, Robert E. Shade, Andrew P. Cap.

**Resources:** Heather F. Pidcoke, Chriselda G. Fedyk, Michael R. Scherer, Robert E. Shade, Andrew P. Cap.

**Software:** James K. Aden.

**Supervision:** Heather F. Pidcoke, Chriselda G. Fedyk, Michael R. Scherer, Robert E. Shade.

**Writing – original draft:** Heather F. Pidcoke, Wilfred Delacruz, Maryanne C. Herzig.

**Writing – review & editing:** Heather F. Pidcoke, Wilfred Delacruz, Maryanne C. Herzig, Beverly S. Schaffer, Sahar T. Leazer, Chriselda G. Fedyk, Robbie K. Montogomery, Nicolas J. Prat, Bijaya K. Parida, James K. Aden, Michael R. Scherer, Robert L. Reddick, Robert E. Shade, Andrew P. Cap.

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
