## [Decision Letter · Decision Letter 0]

24 May 2022

PONE-D-22-06758Perfluorocarbons cause thrombocytopenia, changes in RBC morphology and death in a baboon model of systemic inflammation.PLOS ONE

Dear Dr. Herzig,

Thank you for submitting your manuscript to PLOS ONE. After careful consideration, we feel that it has merit but does not fully meet PLOS ONE’s publication criteria as it currently stands. Therefore, we invite you to submit a revised version of the manuscript that addresses the points raised during the review process.

We look forward to receiving your revised manuscript.

Kind regards,

Arijit Biswas

Academic Editor

PLOS ONE

Journal Requirements:

“This study was funded by the former Oxygen Biotherapeutics Inc., Morrisville, NC. Additional funding was provided by the US Army Medical Research and Materiel Command.”

“Dr. Robert Shade received research funding from Oxygen Biotherapeutics, Inc. The funders had no role in study design, data collection and analysis, decision to publish, or preparation of the manuscript.”

Reviewers' comments:

Reviewer's Responses to Questions

**Comments to the Author**

1. Is the manuscript technically sound, and do the data support the conclusions?

Reviewer #1: Yes

Reviewer #2: Yes

Reviewer #3: Yes

2. Has the statistical analysis been performed appropriately and rigorously? 

Reviewer #1: Yes

Reviewer #2: Yes

Reviewer #3: Yes

3. Have the authors made all data underlying the findings in their manuscript fully available?

Reviewer #1: Yes

Reviewer #2: No

Reviewer #3: Yes

4. Is the manuscript presented in an intelligible fashion and written in standard English?

Reviewer #1: Yes

Reviewer #2: Yes

Reviewer #3: Yes

5. Review Comments to the Author

Reviewer #1: The paper by Pidcoke et al describes a study on the effect of perfluorocarbons on platelet count, RBC morphology, and certain hemostasis parameters in a baboon model of systemic inflammation.

The topic of the paper is of interest. The study, despite its difficulties, has been conducted in an accurate manner and results seem technically and scientifically sound. The materials and methods section of the manuscript is well-detailed. Conclusions drawn from the results are adequate and are fully supported by data. The manuscript is well written.

This reviewer has only one minor comment:

1/ Platelet aggregation results were normalized to platelet count (Figure 4). Although this is reasonable, it is difficult to understand the results of Fig 4B, C and D. One would expect that after such "normalization", baseline value of each group would be 1, which then, of course would change over time depending on the results of platelet aggregation/platelet count. However, on these panels baseline aggregation is close to zero (Fig4B and D) or is between 0.1-0.3. In case the results of this Figure describe only a ratio between maximal aggregation and platelet count, perhaps this should not be referred to as "normalization" to avoid such confusion.

Reviewer #2: A manuscript entitled “Perfluorocarbons cause thrombocytopenia, changes in RBC morphology and death in a baboon model of systemic inflammation” has been submitted for consideration to Plos One

The authors investigated the perfluorocarbon effects on platelet (PLT) function and hemostasis in the lipopolysaccharide (LPS) model of inflammation in the baboon.

The authors concluded that perfluorocarbon infusion caused clinically significant thrombocytopenia and exacerbated LPS-induced platelet activation. The interaction between these effects resulted in decreased hemostatic capacity, diffuse bleeding, shock, and death.

This is an interesting paper with a great animal model and a critical point on blood transfusions.

• The major Comments concerns

1. The manuscript is well written but the introduction is too long and could be more focused on the scientific hypothesis.

2. Please added reference to validate all the methods

3. Please added the clone of each antibody

4. Please added the justification concerning the choice of cytokine/chemokine for inflammatory response characterization.

5. The authors must insert an "Expert Opinion" in the discussion section with a projection into the future of these results with a risk-taking of the authors' experts.

Reviewer #3: The authors explored PFC effects on platelet function and hemostasis in a lipopolysaccharide (LPS) model of inflammation in the baboon. It was demonstrated that PFC infusion caused clinically significant thrombocytopenia and exacerbated LPS-induced platelet activation. The study was interesting and give us an explanation of the PFC caused thrombocytopenia. Yet, there are still some questions need to be answered to make the study more credible.

1.As LPS and PFC caused thrombocytopenia, platelet aggregometry should be tested using washed platelets instead of whole blood. Although the authors normalized platelet accounts.

2.From the manuscript, it’s not clarified if the effect of PFC to increase platelet aggregation was related to PFC itself or its catabolic product. So the authors need to supplement the test of PFC on platelet function in vitro in the presence of and absence of LPS. It will be much better if the authors can explore the signaling pathway changed in platelet after the incubation.

3.Representative curves of platelet aggregation should be given besides of the statistical data to make the results more visualized.

4.As the authors explained the thrombocytopenia as the increasing activation and clearance of platelet, the flow cytometry with biomarker of microphage and the classification of circulating platelets such as the constitution of pre-mature and mature platelet should be given.

5.As shown in Line 282 in the version of the article, the combination of LPS and PFC caused platelet activation similar to observations in the LPS-SAL group. Yet from Figure 4, the combination of LPS and PFC caused platelet activation much closer to observations in the SAL-SAL group. So the author need to verify the results.

6.Some of the icons in the text is not in accordance with that in the Figure, such as the Figure 7D. The authors should correct the mistakes carefully.

6. PLOS authors have the option to publish the peer review history of their article (what does this mean?). If published, this will include your full peer review and any attached files.

Reviewer #1: No

Reviewer #2: No

Reviewer #3: No

---

## [Author Response · Author response to Decision Letter 0]

22 Nov 2022

Reviewer #1: The paper by Pidcoke et al describes a study on the effect of perfluorocarbons on platelet count, RBC morphology, and certain hemostasis parameters in a baboon model of systemic inflammation. The topic of the paper is of interest. The study, despite its difficulties, has been conducted in an accurate manner and results seem technically and scientifically sound. The materials and methods section of the manuscript is well detailed. Conclusions drawn from the results are adequate and are fully supported by data. The manuscript is well written. This reviewer has only one minor comment:

1/ Platelet aggregation results were normalized to platelet count (Figure 4). Although this is reasonable, it is difficult to understand the results of Fig 4B, C and D. One would expect that after such "normalization", baseline value of each group would be 1, which then, of course would change over time depending on the results of platelet aggregation/platelet count. However, on these panels baseline aggregation is close to zero (Fig4B and D) or is between 0.1-0.3. In case the results of this Figure describe only a ratio between maximal aggregation and platelet count, perhaps this should not be referred to as "normalization" to avoid such confusion. 

The data is expressed not as relative to baseline values, which would then have baseline numbers of 1, as mentioned by the reviewer, but are expressed as the platelet aggregation per platelet in the sample. With high platelet numbers, this accounts for aggregation per platelet values of near zero. To clarify this, the text on page 14 has been changed to read: 

There is a pronounced thrombocytopenia with PFC with a75-80% drop in platelet number by 72h (see Fig 3) and as thrombocytopenia may affect aggregometry results, masking PFC and LPS effects on agonist response, the data was normalized to PLT count at each time point (Fig 4b). lines288-291

Reviewer #2: A manuscript entitled “Perfluorocarbons cause thrombocytopenia, changes in RBC morphology and death in a baboon model of systemic inflammation” has been submitted for consideration to Plos One The authors investigated the perfluorocarbon effects on platelet (PLT) function and hemostasis in the lipopolysaccharide (LPS) model of inflammation in the baboon. The authors concluded that perfluorocarbon infusion caused clinically significant thrombocytopenia and exacerbated LPS-induced platelet activation. The interaction between these effects resulted in decreased hemostatic capacity, diffuse bleeding, shock, and death. This is an interesting paper with a great animal model and a critical point on blood transfusions.

• The major Comments concerns

1. The manuscript is well written but the introduction is too long and could be more focused on the scientific hypothesis. 

The introduction has been trimmed in several parts, (tracked lines 64-65; 73-76; 87; 99). The scientific hypothesis has been expanded and now reads (lines 93-99): Baboons have historically been used as a model for human platelet function (28) and Oxygen Biotherapeutics, evaluated the maximum tolerated dose (MTD) of Oxycyte and its effects in baboons at the Texas Biomedical Research Institute (San Antonio, Texas). Baboons develop sepsis in response to lipopolysaccharides of gram-negative bacteria with manifestation of acute cytokine responses and thrombocytopenia (Haudek 2003). Therefore a lipopolysaccharide (LPS) model of inflammation in the baboon was used in this study in conjunction with the PFC Oxycyte to provide This study provided a comprehensive evaluation of the PFC Oxycyte’s effects on platelet function and hemostasis under conditions of exacerbated thrombocytopenia.in a lipopolysaccharide (LPS) model of inflammation in the baboon.

2. Please added reference to validate all the methods 

References for flow cytometry (Schnizlein-Bick, 2000), TEG and multiplate analyses (Bochsen, 2011) have been added to the manuscript in positions (lines 190, 199 and 208).

3. Please added the clone of each antibody

The clones of the antibodies have been added when available. Lines 174-181 now read:

10µl PFP was added to an antibody cocktail containing the following antibodies: APC labeled CD41a, clone HIP8; PE labeled CD62P, clone AC1.2; and V450 labeled CD45, clone D058-1283 (all from BD Biosciences, San Jose, California, USA,); PerCP-Cy5.5 labeled CD144 (Santa Cruz Biotechnology, Dallas, Texas, USA,) and 100 nM FITC labeled lactadherin (Haematologic Technologies, Inc., Essex Junction, Vermont, USA,). Isotype controls were used for non-specific fluorescence: PE mouse IgG1k, clone x40; PerCP isotype, mouse igG 1K, clone MOPC-21; APC isotype, Mouse igG 1K, clone MOPC-21; V450 isotype control clone MOPC-21. 

4. Please added the justification concerning the choice of cytokine/chemokine for inflammatory response characterization. 

The introduction now introduces the inflammatory response info:

Line 87-89: Baboons develop sepsis in response to lipopolysaccharides of gram-negative bacteria with manifestation of acute cytokine responses and thrombocytopenia {Haudek, 2003 }. 

Further justification is in the Results section, line 248. 

LPS is known to elicit acute responses in the baboon in interleukin 6 (IL_6), tumor necrosis factor alpha (TNF�), macrophage inflammatory proteins 1, alpha and 1 beta (MIP1�, MIP1�) as well as interleukin 1 receptor antagonist (IL-1Ra) (Haudek, 2003, Angelova, 2004).

5. The authors must insert an "Expert Opinion" in the discussion section with a projection into the future of these results with a risk-taking of the authors' experts. 

Line 437 in Discussion begins the expert opinion.

Expert Opinion: This study emphasizes the critical importance of attempting to anticipate clinical safety issues in the interrelationships of trauma, hemorrhage, and sepsis and then devising studies to test critical physiologic pathways during pre-clinical product development. This study identified thrombocytopenia and lethal hemorrhage as an adverse safety concern when the PFC blood substitute, Oxycyte, was administered in a baboon non-human primate model in the setting of LPS-induced inflammation. The findings obtained in this clinically relevant model contributed to termination of development efforts and avoided potential serious adverse events in human subjects. 

Reviewer #3: The authors explored PFC effects on platelet function and hemostasis in a lipopolysaccharide (LPS) model of inflammation in the baboon. It was demonstrated that PFC infusion caused clinically significant thrombocytopenia and exacerbated LPS-induced platelet activation. The study was interesting and give us an explanation of the PFC caused thrombocytopenia. Yet, there are still some questions need to be answered to make the study more credible.

1. As LPS and PFC caused thrombocytopenia, platelet aggregometry should be tested using washed platelets instead of whole blood. Although the authors normalized platelet accounts.

As the reviewer stated, the data was normalized to platelet counts. We feel that this is sufficient to address the concerns.

2. From the manuscript, it’s not clarified if the effect of PFC to increase platelet aggregation was related to PFC itself or its catabolic product. So the authors need to supplement the test of PFC on platelet function in vitro in the presence of and absence of LPS. It will be much better if the authors can explore the signaling pathway changed in platelet after the incubation.

While that would be an interesting study, it is beyond the scope of this study which was to address whether this drug would be safe in an already compromised thrombocytopenic patient. Whether due to the PFC itself or a catabolic product, the baboons died. As to whether a catabolic product is responsible, in the literature are reports that fluorocarbons are “among the most stable, most inert chemicals known to man” Riess, 2009. In any event, we are unable to acquire any more reagent even if we sought to test this question; the company folded after the results of this study. However, it is unlikely that the signaling pathway in platelets is changed; more likely is that it is overwhelmed by the presence of thrombogenic PFC particles.

3. Representative curves of platelet aggregation should be given besides of the statistical data to make the results more visualized.

New Supplemental Figure S4 gives screenshot raw data of the representative platelet aggregation.

4. As the authors explained the thrombocytopenia as the increasing activation and clearance of platelet, the flow cytometry with biomarker of microphage and the classification of circulating platelets such as the constitution of pre-mature and mature platelet should be given. 

In re-examining the data, the mean platelet volume (mpv) revealed that while platelet numbers decreased, remaining platelets became smaller and denser, indicative of older mature platelets (Korniluk 2019). When platelet numbers increased, the mpv significantly increased, indicative of younger platelets. This data is now shown in supplemental figure S3. We did use flow cytometry analysis of platelets with thiazole orange staining to determine early platelets, Kienast 1990, however the data had some missing samples and only showed trends with no significant differences of CD41+/THOR+ population and was not included in the paper. That data however is shown below.

The text at line 269 has been modified to read: The thrombocytopenia observed correlates with decreased mean platelet volume (MPV) followed by an increasing MPV, indicative of younger platelet populations (Figure S3). 

5. As shown in Line 282 in the version of the article, the combination of LPS and PFC caused platelet activation similar to observations in the LPS-SAL group. Yet from Figure 4, the combination of LPS and PFC caused platelet activation much closer to observations in the SAL-SAL group. So the author need to verify the results.

We stand by our original statement that LPS-PFC is more similar to LPS-SAL than SAL-SAL. In Figure 4A, infusion of LPS causes an initial drop in activation followed by an increase in activation for both LPS-SAL and LPS-PFC. This is in opposition to SAL-SAL where the activation is steady. When examined on a per platelet basis as seen in Figure 4B, the similarity is more clear; in this analysis, LPS-PFC3 is shown to be even more active on a per-platelet basis than LPS-SAL; SAL-SAL values are near baseline values. 

6. Some of the icons in the text is not in accordance with that in the Figure, such as the Figure 7D. The authors should correct the mistakes carefully.

We are grateful for the reviewers careful reading of the text. The designations have been changed in the text to match the figure (Fig 7C changed to Fig 7A; Fig 7D changed to Fig 7B; line 351 tracked). Additionally, in a careful examination of figure legends and text, other errors were noted and corrected. Figure 5 legend had OXY replaced with PFC as needed (lines 329-332 tracked). Figure 6 had the graphs re-arranged to match the legends and texts (new Fig 6).

Additional concerns were noted in “Journal requirements”.

Journal Requirements:

1. Please ensure that your manuscript meets PLOS ONE's style requirements, including those for file naming. The PLOS ONE style templates can be found at Caution-https://journals.plos.org/plosone/s/file?id=wjVg/PLOSOne_formatting_sample_main_body.pdf and Caution-https://journals.plos.org/plosone/s/file?id=ba62/PLOSOne_formatting_sample_title_authors_affiliations.pdf

< Caution-https://journals.plos.org/plosone/s/file?

id=ba62/PLOSOne_formatting_sample_title_authors_affiliations.pdf >

The reference to funding in the body of the text has been removed. Our amended funding statement is below.

3. In your Data Availability statement, you have not specified where the minimal data set underlying the results described in your manuscript can be found. PLOS defines a study's minimal data set as the underlying data used to reach the conclusions drawn in the manuscript and any additional data required to replicate the reported study findings in their entirety. All PLOS journals require that the minimal data set be made fully available. For more

information about our data policy, please see Caution-http://journals.plos.org/plosone/s/data-availability.

Upon re-submitting your revised manuscript, please upload your study’s minimal underlying data set as either Supporting Information files or to a stable, public repository and include the relevant URLs, DOIs, or accession numbers within your revised cover letter. For a list of acceptable repositories, please see Caution-http://journals.plos.org/plosone/s/data-availability#loc-recommended-repositories. Any potentially identifying patient information must be fully anonymized. Important: If there are ethical or legal restrictions to sharing your data publicly, please explain these restrictions in detail. Please see our guidelines for more information on what we consider unacceptable restrictions to publicly sharing data: Caution-http://journals.plos.org/plosone/s/data-availability#loc-unacceptable-data-access-restrictions.

Note that it is not acceptable for the authors to be the sole named individuals responsible for ensuring data access. We will update your Data Availability statement to reflect the information you provide in your cover letter.

The underlying data for this study has been collected into an excel file. No ethical or legal restrictions exist. The excel file is: PLOS_PFC_Data

“This study was funded by the former Oxygen Biotherapeutics Inc., Morrisville, NC. Additional funding was provided by the US Army Medical Research and Materiel Command.”

We note that you have provided additional information within the Acknowledgements Section that is not currently declared in your Funding Statement. Please note that funding information should not appear in the Acknowledgments section or other areas of your manuscript. We will only publish funding information present in the Funding Statement section of the online submission form. Please remove any funding-related text from the manuscript and let us know how you would like to update your Funding Statement. Currently, your Funding Statement reads as follows:

“Dr. Robert Shade received research funding from Oxygen Biotherapeutics, Inc. The funders had no role in study design, data collection and analysis, decision to publish, or preparation of the manuscript.”

The reference to funding in the body of the text has been removed. Our amended funding statement should read as follows: 

Dr. Robert Shade received research funding from Oxygen Biotherapeutics, Inc. The funders had no role in study design, data collection and analysis, decision to publish, or preparation of the manuscript. Additional funding was provided by the US Army Medical Research and Materiel Command.”

Ethics statements constrained to Methods section.

---

## [Editor Report · Decision Letter 1]

13 Dec 2022

Perfluorocarbons cause thrombocytopenia, changes in RBC morphology and death in a baboon model of systemic inflammation

PONE-D-22-06758R1

Dear Dr. Herzig,

We’re pleased to inform you that your manuscript has been judged scientifically suitable for publication and will be formally accepted for publication once it meets all outstanding technical requirements.

Kind regards,

Arijit Biswas

Academic Editor

PLOS ONE

---

## [Editor Report · Acceptance letter]

19 Dec 2022

PONE-D-22-06758R1 

Perfluorocarbons cause thrombocytopenia, changes in RBC morphology and death in a baboon model of systemic inflammation 

Dear Dr. Herzig:

I'm pleased to inform you that your manuscript has been deemed suitable for publication in PLOS ONE. Congratulations! Your manuscript is now with our production department. 

Kind regards, 

on behalf of

Dr. Arijit Biswas 

Academic Editor

PLOS ONE